# Flipping antimicrobial peptides in the exit tunnel of the bacterial ribosome

Weiping Huang[1,2,4,6], Max J. Berger [3,6], Haaris A. Safdari [3], Dorota Klepacki[1,2], Helge Paternoga [3], Chetana Baliga [1,2,5], Daniel N. Wilson [3] ✉, Nora Vázquez-Laslop [1,2] ✉ & Alexander S. Mankin [1,2] ✉

Proline-rich antimicrobial peptides (PrAMPs) kill bacteria by binding in the ribosomal nascent peptide exit tunnel. Type II PrAMPs bind in an orientation matching that of the nascent protein, trap the release factors and arrest ribosomes at stop codons. Conversely, Type I PrAMPs bind in an opposite orientation: their N-terminus invades the peptidyl transferase center arresting translation at start codons. Here, by mining the genome databases, we identify a number of PrAMPs with high sequence similarity to the Type II PrAMP Drosocin. Notably, many of the new PrAMPs do not stall ribosomes at stop codons, but act as Type I PrAMPs arresting translation at start codons. Structural analysis shows that such peptides bind with a Type I orientation. Minimal alterations in the peptide structure can flip the orientation of the PrAMP in the exit tunnel, switching the mechanism of translation inhibition. Altering the mode of binding and action of a PrAMP by only few mutations could be exploited by the host to combat newly emerging bacterial pathogens.

The ribosome synthesizes proteins by interpreting the genetic information encoded in mRNA. Aminoacyl-tRNAs are selected according to the sequence of the mRNA codons and amino acids are polymerized into polypeptides in the catalytic peptidyl transferase center (PTC) of the large (50S) ribosomal subunit. As new amino acids are added one by one to the C-terminus of the protein, the growing polypeptide is extruded, N-terminus-first, through the ~100 Å long and 20-30 Å wide nascent polypeptide exit tunnel (NPET), which spans the body of the large subunit[1]. While passing through the NPET, the nascent polypeptide forms transient interactions with diverse ribosomal RNA (rRNA) and protein tunnel elements. This co-translational interplay shapes the overall trajectory and the local conformation of the nascent chain in the NPET and may significantly influence PTC activity, as well as the folding, targeting and release of the newly synthesized proteins[2–8]. However, nascent proteins are not the only polypeptides which are able to interact with the NPET as they are being synthesized; some proteins can do so even post-translationally. For example, several

proteins involved in ribosome biogenesis, dormancy, or protein targeting operate by inserting their N- or C-terminal tails into the NPET through its outlet, reaching in some cases right up to the PTC[9–14], while others can stretch into the NPET from the PTC side inlet[15,16].

A special place among the proteins that bind post-translationally in the NPET belongs to the ribosome-targeting proline-rich antimicrobial peptides (PrAMPs). Encoded in the genomes of mammals and insects, PrAMPs are components of the innate immune system providing defense against bacterial infections[17,18]. Due to their small size (usually, less than 30 amino acid residues), PrAMPs can rapidly evolve to control growth of new harmful bacteria playing an important role in the evolutionary race between pathogens and the host[19]. The diversity, potency, malleability and low cytotoxicity of PrAMPs make them attractive prototypes for developing clinically useful antibiotics that are able to combat drug-resistant pathogens[20–24].

PrAMPs kill bacteria by acting upon the ribosome[25–27]. They enter bacterial cells by hijacking specific peptide transporters[28–30] and once

[1]Department for Pharmaceutical Sciences, University of Illinois, Chicago, IL, USA. [2]Center for Biomolecular Sciences, University of Illinois, Chicago, IL, USA. [3]Institute for Biochemistry and Molecular Biology, University of Hamburg, Hamburg, Germany. [4]Present address: College of Agronomy and Biotechnology, Yunnan Agricultural University, Kunming, China. [5]Present address: Faculty of Life and Allied Health Sciences, Department of Biotechnology, M. S. Ramaiah University of Applied Sciences, Bangalore, India. [6]These authors contributed equally: Weiping Huang, Max J. Berger. ✉e-mail: daniel.wilson@uni-hamburg.de; nvazquez@uic.edu; shura@uic.edu

in the cytoplasm, enter the ribosomal NPET[25–27,31–38]. Because the NPET accommodates just a single polypeptide chain at any given time, PrAMPs can bind only when the tunnel is vacant: either when the ribosome is positioned at a start codon and has not yet begun elongation, or after it reaches a stop codon and releases the completed protein[39].

The known PrAMPs differ dramatically in their mode of binding and action. Type I PrAMPs such as, for example, the 19-amino acid long oncocin112 (Onc112) or the 20-residue-long pyrrhocoricin (Pyr), bind in an orientation opposite to that of the growing polypeptide, with the C-terminus extending down the NPET and the N-terminus invading the A site of the PTC (Fig. 1a, b)[31–34]. As a result, Type I PrAMPs preclude binding of the first elongator aminoacyl-tRNA in the A site and arrest the initiating ribosome at the start codon[22,31–34,40]. The binding mode and mechanism of action of Type II PrAMPs, exemplified by 19- and 18-residues long drosocin (Dro) and apidaecin (Api), respectively, is contrastingly different. Unlike Type I PrAMPs, Dro and Api bind in the NPET in the orientation matching that of the growing polypeptide, that is, with the N-terminus protruding towards the NPET exit, and the C-terminus approaching the PTC (Fig. 1c, d)[35,36,41]. In stark contrast to the action of Type I PrAMPs, Type II PrAMPs arrest the ribosome not at the start codon but at the stop codon. They enter the NPET after the nascent protein is released, diffuse towards the PTC, and trap the ribosome-associated class I release factors (RF1 or RF2) due to the interaction of the PrAMP's penultimate arginine residue with the glutamine of the conserved GGQ motif of class I RFs. As a result, Type II PrAMPs trap RF1/RF2 on the post-release ribosome preventing their dissociation needed for the completion of the termination stage of translation[35–37,42,43]. Furthermore, because in bacteria ribosomes are present in excess over RFs, depletion of RFs in the Type II PrAMP-treated cells result in stalling of the majority of the cellular ribosomes at the stop codons in a pre-release stage revealing these PrAMPs as bona fide termination inhibitors[35,43].

Despite their disparate mechanisms of action, Type I and Type II PrAMPs have rather similar sequences. The amino acid composition of both types of PrAMPs is characterized by a high content of proline and arginine residues, some of which are arranged in conserved sequence motifs[39,41,44]. Such similarity presents a significant challenge for predicting the mode of PrAMP action based solely on sequence analysis. Thus, for example, only after detailed structural and biochemical analyses was Dro established as a Type II PrAMP[36,37]. Uncertainty in assigning the mode of action of the PrAMPs as either Type I or Type II based solely on the amino acid sequence likely applies to other newly found and under-studied PrAMPs[44].

In this work, to explore the diversity of ribosome-targeting antimicrobial peptides, we searched insect genomes for genes encoding Dro-like PrAMPs and found them in fly species from two subgenera of the genus *Drosophila*. We find that while the PrAMPs of the subgenus *Sophophora* exhibit typical Type II PrAMP characteristics, the Dro-like PrAMPs from the subgenus *Drosophila* display all the hallmarks of Type I PrAMPs: they bind to the NPET and PTC in the inverted orientation and inhibit translation initiation, rather than termination. Subsequent biochemical and structural analyses of engineered chimeras of Type I and Type II Dro-like PrAMPs show that minimal alterations in the amino acid sequence, sometimes addition, deletion, or substitution of a single amino acid residue, is sufficient to flip the orientation of the peptide in the NPET and change its mode of translation inhibition. These findings show the general propensity of the arginine- and proline-rich sequences for interacting with the NPET in either orientation and illuminate a potential mechanism for fine-tuning the mode of PrAMPs' action by flipping the direction of its placement in the NPET. Our findings demonstrate a strategy that may be leveraged by the host to combat emerging bacterial pathogens.

## Results
### Identification of new Dro-like PrAMPs
Dro is encoded in the genome of *Drosophila melanogaster* as a pre-proprotein whose processing generates the mature 19-amino acid long PrAMP with the sequence GKPRPYSPRPTSHPRPIRV. To identify new Dro-like PrAMPs, we interrogated genome databases using the sequences of the mature Dro and its pre-pro-protein as probes. In agreement with previous studies that analyzed a then much smaller available database[45], we found genes of Dro-like PrAMPs in the fly species belonging to two subgenera of the genus *Drosophila*. The genomes of species of the subgenus *Sophophora* carry a single copy gene that encodes a pre-pro-protein encompassing the sequence of the mature PrAMP. Additionally, we found species of the subgenus *Drosophila* carrying either multiple genes (2–3 per genome) specifying unique Dro-like peptides, or a single gene with multiple (3–10) tandem repeats/isoforms of mature Dro-like peptides separated by protease cleavage sites (Fig. 2).

In total, we found 26 unique Dro-like PrAMPs derived from seven *Sophophora* and nine *Drosophila* species (Fig. 3a, Supplementary Tables 1, 2). The identified peptides are 18–19 amino acids long, rich in proline and arginine residues and show high sequence similarity to Dro and between each other (Fig. 3a). All peptides share the tetrapeptide PRPT motif in the middle of their sequence. In addition, the identified peptides carry a highly conserved N-terminal glycine, a hydrophobic C-terminal residue, and K/R-P motifs in their N- and C-terminal

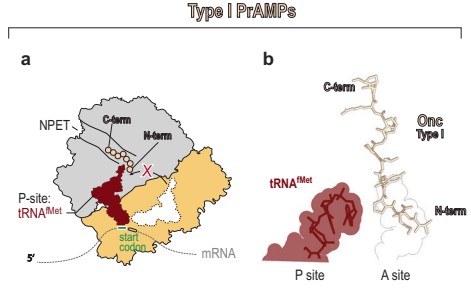

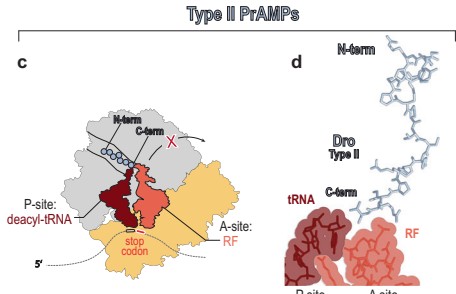

**Fig. 1 | Type I and Type II PrAMPs bind to the ribosome with opposite orientations to inhibit translation via disparate mechanisms. a** Type I PrAMPs bind in the vacant nascent peptide exit tunnel (NPET) of the initiating ribosome in an orientation opposite to that of the growing protein chain. The N-terminus of a Type I PrAMP blocks accommodation of the acceptor end of the first elongator aminoacyl-tRNA in the A site of the peptidyl transferase center (PTC), arresting the ribosome at start codons. **b** Binding of the Type I PrAMP oncocin (Onc) (PDB ID 5HCR)[34] in the NPET. **c** Type II PrAMPs bind in the NPET of the post-release ribosome in an orientation matching that of the growing protein chain. The C-terminus of a Type II PrAMP approaches the PTC and traps the P-site bound deacylated tRNA and the A-site associated release factor (RF1 or RF2) preventing RF dissociation and thereby arresting the ribosome at the stop codon. **d** Binding of the Type II PrAMP drosocin (Dro)(PDB ID 8AKN)[36] in the NPET.

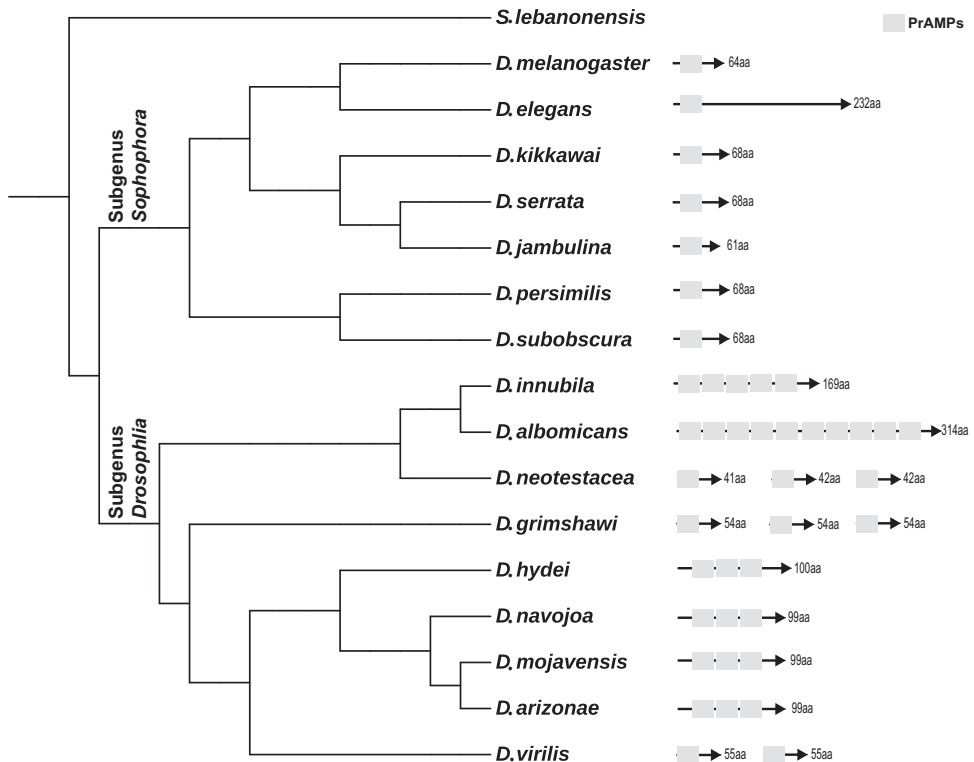

**Fig. 2 | Organization of genes encoding Dro-like PrAMPs in fly genomes.** The number of amino acids in the predicted pre-pro-protein is indicated. The number of PrAMP repeats in each protein is shown by gray boxes. Genome-based phylogenetic tree reconstruction is from ref. 73.

segments. Notably, the penultimate C-terminal residue in all peptides is arginine, the crucial amino acid for the interaction of Type II PrAMPs with the ribosome-bound RFs[35,36,41]. The scattered distribution of conserved elements across the entire PrAMP sequence aligns well with the previous findings showing that residues critical for binding of Dro in the NPET of the terminating ribosome are dispersed through the entire length of the peptide[37].

## Dro-like PrAMPs exhibit contrasting modes of action

The defining functional feature of Type II PrAMPs, including Dro, is trapping ribosomes after the release of the completed protein[35–37,43,46]. Therefore, we selected a few of the newly identified Dro-like PrAMPs (two peptides from the subgenus *Sophophora* with single amino acid changes compared to Dro and seven peptides from the subgenus *Drosophila* deviating from the Dro sequence by several residues) (Fig. 3a) to chemically synthesize them and interrogate their ability to arrest ribosomes at stop codons (Supplementary Table 3). To achieve this, we employed toeprinting analysis, a technique that enables precise mapping of the mRNA position of a ribosome stalled during in vitro translation[47–49]. Similar to Dro, two PrAMPs, Dse and Del from the subgenus *Sophophora*, arrested ribosomes at the stop codon of a model mRNA, as expected for genuine Type II PrAMPs (Fig. 3b). Unexpectedly, however, the seven tested PrAMPs from the subgenus *Drosophila* (Dal1, Dal2, Dal3, Dhy1, Din1, Din2, and Dna1) stalled ribosomes at the start codon akin to the mode of action of Type I PrAMPs (Fig. 3b).

By arresting the initiating ribosome, Type I PrAMPs are known to readily interfere with the in vitro translation of a reporter protein. In contrast, Type II PrAMPs are notably poor inhibitors of in vitro protein synthesis because they bind to the terminating ribosome only after the completion of the first round of translation and exert their inhibitory action only after RFs are exhausted in a cell-free translation reaction[25,36,37]. Consistent with their Type I-like behavior in the toeprinting assay, Dal1, Dal2, Dal3, Dhy1, Din1, Din2, and Dna1 PrAMPs,

strongly inhibited cell-free translation of GFP, whereas Dro, Dse and Del, determined to be Type II PrAMPs by the same approach, only marginally decreased expression of the reporter (Fig. 3c).

Taken together, our in vitro experiments showed that minor sequence differences among highly similar Dro-like PrAMPs encoded in fly genomes lead to their highly contrasting effects on translation.

## Dal2 binds to the ribosome as a Type I PrAMP

Different scenarios could account for the ribosome arrest at the start codon caused by Dal2 and other Dro-like PrAMPs from the subgenus *Drosophila*. Our previous cryo-EM analysis of the ribosome complexed with Dro showed a minor subpopulation of particles with the PrAMP in the NPET in the classic Type II orientation (Fig. 1c, d) but bound to the initiating, not the terminating, ribosome, presumably inhibiting the first act of translocation[36]. Thus, in principle, Dal2 and its siblings could act as early translocation inhibitors. Another possibility is that while these peptides bind in the NPET still maintaining the Type II PrAMP orientation (Fig. 1c, d), their C-terminal residues could be slightly shifted towards the PTC active site, similar to what was observed with a C-terminally modified Api[38]; even a minimal shift of the PrAMP's C-terminus towards the PTC would create a clash with the incoming aminoacyl-tRNA leading to ribosome stalling at the start codon. Yet a third possibility that cannot be excluded is that the few amino acid substitutions that differentiate subgenus *Drosophila* Dro-like PrAMPs from the *Sophophora* Type II PrAMPs are sufficient to flip the peptides' orientation in the ribosomal NPET, allowing them to bind as a Type I.

To distinguish between these possibilities, we determined a cryo-EM structure of a Dal2-stalled ribosome complex (Dal2-SRC). The Dal2-SRC was prepared by in vitro translation of a short MLIF protein in the presence of Dal2, as used previously for preparing Dro-SRC for cryo-EM analysis[36]. In silico sorting of the cryo-EM data revealed one major population of ribosomes with a tRNA bound in the P-site and additional density for the Dal2 peptide within the NPET (Fig. 4a, Supplementary Figs. 1–4), which could be refined to an average resolution of

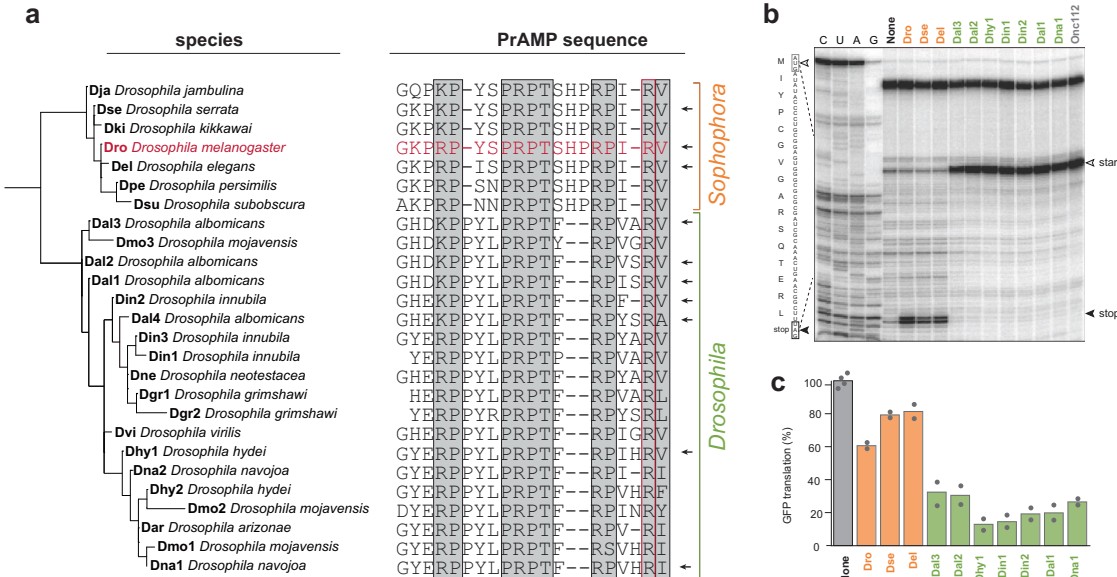

**Fig. 3 | Contrasting activities of Dro-like PrAMPs encoded in fly genomes.**
**a** Similarity tree and sequence comparison of the Dro-like PrAMPs identified in the genomes of flies from subgenera *Sophophora* and *Drosophila* of the genus *Drosophila*. The sequence of Dro is shown in red. PrAMPs were named using a 3-character species designation of the producer, followed by a number when several PrAMP isoforms were encoded in the same genome. The universally conserved sequences are highlighted by gray boxes and the penultimate arginine residue, which is critical in Dro for trapping the RF[37], is boxed in red. The peptides that were chemically synthesized and experimentally tested are indicated by arrowheads. **b** Toeprinting analysis of the mode of action of Dro-like peptides. A model mini ORF derived from the modified *E. coli yrbA* gene was translated in vitro in the absence (None) or presence of PrAMPs from subgenera *Sophophora* (orange) or *Drosophila* (green)

and the site of the PrAMP-induced ribosome arrest was determined by primer extension. The bands corresponding to ribosomes arrested at the start and stop codons are indicated by white and black arrowheads, respectively. Sequencing lanes are indicated and the nucleotide and amino acid sequences of the *yrbA* ORF are shown. The gel is a representative of three independent experiments.
**c** Inhibition of translation of the reporter protein sfGFP in an *E. coli* cell-free translation system by Dro-like PrAMPs from the subgenera *Sophophora* (orange) or *Drosophila* (green). The fluorescence values at the 50 min time point, obtained in two independent experiments, were normalized to the fluorescence values recorded in the control reactions that lacked PrAMPs (None). Dots indicate the results of independent experiments (with duplicate controls in each); the height of the bars represents the experimental mean.

2.5 Å (Supplementary Figs. 1, 2, Supplementary Table 4). Inspection of the density for the P-tRNA, especially the codon-anticodon interaction with the mRNA, indicated that, as expected based on the toeprinting assays (Fig. 3b), it was the initiator tRNA that was positioned at the P-site (Supplementary Fig. 3a, b). The density in the exit tunnel was well-resolved and allowed 13 residues (His2-Arg14) of Dal2 to be modeled (Fig. 4b), whereas the N-terminal Gly1 and five C-terminal residues ($_{15}$PVSRV$_{19}$) were poorly ordered and therefore not included in the model. The density suggests that Lys4 of Dal2 and C2573 of the 23S rRNA are present in two alternative conformations possibly due to a steric clash of the latter with Gly1 of Dal2 that would be predicted to extend in its direction (Supplementary Fig. 3c, d).

Dal2 establishes multiple direct and water-mediated hydrogen bonds (H-bonds) as well as stacking interactions with 23S rRNA residues in the NPET (Supplementary Fig. 4). Its overall conformation and binding mode are similar to those observed previously for the Type I PrAMP Oncocin (Onc) and its derivatives (Fig. 4c)[32,33]. However, compared to Onc, Dal2 forms two additional stacking interactions with 23S rRNA, one between its His2 residue and the nucleobase of U2555 and a second between its Phe13 and U2586 (Supplementary Fig. 4a, d), neither of which is possible for Onc due to the presence of Val and Pro respectively, at the equivalent sequence positions. Despite the difference in the overall conformation of the C-termini of Dal2 and Onc, both peptides form stacking interactions with A2062 but utilizing different arginine sidechains, specifically, Arg14 for Dal2 and Arg11 for Onc (Supplementary Fig. 4a, d). In addition, we also observe a number of water-mediated interactions (Supplementary Fig. 4c) that differ from those reported for Onc (and its derivatives)[32–34].

Analogous to Onc and other Type I PrAMPs[31–34], the N-terminus (His2-Pro5) of Dal2 occludes the space occupied by the CCA-end of aminoacyl-RNA in the ribosomal A site (Fig. 4d). Furthermore, Asp3 of Dal2 mimics the C75 of the tRNA by forming hydrogen bonds with G2553 located in the A-loop of the 23S rRNA (Fig. 4a, d), as observed previously for Onc[32–34]. Additionally, His2 of the Dal2 PrAMP uniquely mimics the C74 base of the A-tRNA by forming a hydrogen bond with U2554 (Fig. 4d), thus presenting an additional obstacle for the incoming aminoacyl-tRNA that is not present in Onc.

The analysis of the Dal2-SRC structure shows that in addition to interference with A-site tRNA accommodation, Dal2 also hinders placement of the fMet-tRNA acceptor end. We note that Tyr7 and Leu8 of Dal2 impinge upon the space occupied by the fMet-tRNA formyl group in the initiating ribosome[50]. Such encroachment and the unusual orientation of U2585 in the Dal2-SRC, displaces fMet from its canonical position (Fig. 4e). Perhaps because of this incompatibility, A76 of the fMet-tRNA adopts an inverted conformation (Fig. 4f) such that the ribose of A76 now faces towards the canonical position of A2602, causing it to flip by 180° (Supplementary Fig. 3e, f). Thus, Dal2, and likely also other Type I PrAMPs, arrest the ribosome at the start codon by simultaneously displacing the aminoacyl end of the P-site initiator tRNA and occluding the binding site of the first elongator tRNA.

In sum, our structural studies have revealed that, regardless of its high sequence similarity to the Type II Dro, Dal2 has all the characteristics of Type I PrAMPs: it adopts an inverted orientation in the ribosome and causes translation arrest not at the stop but at the start codon.

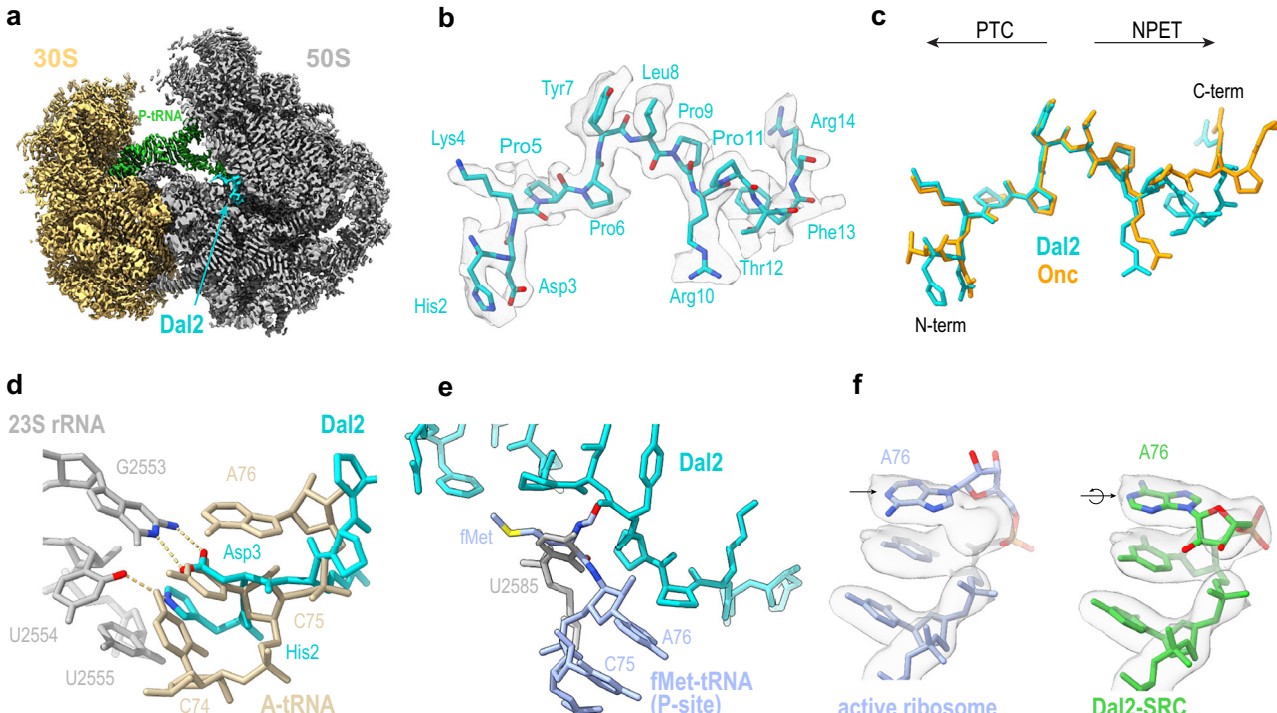

**Fig. 4 | Cryo-EM structure of Dal2-stalled ribosome complex. a** Transverse section of the cryo-EM map of the Dal2-SRC (30S, yellow; 50S gray) with P-tRNA (green) and Dal2 (cyan) shown. **b** Cryo-EM map density (transparent gray surface) and molecular model (cyan) of residues His2-Arg14 of Dal2 bound within the Dal2-SRC. **c** Comparison of binding mode of Dal2 (cyan) and Onc112 (orange) based on alignment of the Dal2-SRC with the Onc112-ribosome complex (PDB ID 5HCR)[34].

**d** Clash between N-terminal residues of Dal2 (cyan) and CCA-end of A-site tRNA (beige). **e** Overlay of Dal2 and U2585 from the Dal2-SRC with the position of the ribosome-bound fMet-tRNA (PDB ID 1VY4)[50]. **f** Comparison of the cryo-EM densities of the CCA-end of the P-site tRNA in the active ribosome (PDB ID 7K00)[64] (left) and of the P-site tRNA in the Dal2-SRC (right). The rotation of the A76 base is indicated by an arrow.

## Changing the N-terminal segment of Dro is sufficient to flip its orientation in the ribosome

Intrigued by the contrasting modes of binding and action of native PrAMPs with high sequence similarity, we asked how small the sequence changes could be that would reverse the peptide's orientation in the NPET and alter its mode of action. For the initial experiments, we chose two peptides, Dro and Dal2 (Fig. 5a). Dro is a typical Type II PrAMP that binds in the orientation matching that of the growing nascent polypeptide chain (Fig. 1c, d)[36] and arrests the terminating ribosome by trapping the RFs (Fig. 3b). Conversely, Dal2 binds as a Type I PrAMP - in an inverted orientation, with the N-terminus invading the PTC and the C-terminus protruding into the NPET (Fig. 4a, c), and arrests the ribosome at the start codon (Fig. 3b). Both PrAMPs contain the identical tetrapeptide sequence PRPT in the middle, but the segments flanking this motif vary, even if remaining generally similar (Fig. 5a).

We first explored whether it is the N- or the C-terminal sequences that drive the mode of binding and action of the PrAMP. To do so, we constructed two chimeras. In DroDal, the C-terminal segment of Dro adjacent to the PRPT motif was replaced with that of Dal2, whereas in DalDro, the N-terminal segment of Dro was substituted with the corresponding sequence from Dal2 (Fig. 5a). Toeprinting analysis showed that while DroDal was inactive, the DalDro peptide behaved as a typical Type I PrAMP, arresting the ribosome at the start codon (Fig. 5b). Thus, replacing the N-terminal segment of Dro with the corresponding Dal2 sequence dramatically shifted its mechanism of action from that characteristic of Type II PrAMPs to that of Type I.

To find out how DalDro binds to the ribosome, we determined a cryo-EM structure of a DalDro-SRC (Fig. 5, Supplementary Figs 5–8, Supplementary Table 4). As for the Dal2-SRC, the DalDro-SRC was

prepared by in vitro translation of the MLIF protein in the presence of DalDro and then subjected to single particle cryo-EM analysis. In silico sorting of the cryo-EM data revealed two main populations of ribosomes, with the major one (73% of the 70S particles) containing an initiator tRNA bound in the P-site and additional density for the DalDro peptide bound in the inverted (Type I) orientation within the NPET (Fig. 5c, d, Supplementary Figs. 5, 7), which could be refined to an average resolution of 2.9 Å (Supplementary Fig. 6). The density in the exit tunnel was sufficiently well-resolved to allow 16 residues (Gly1-Arg16) of DalDro to be modeled (Fig. 5d). The conformation and interactions of residues His2-Thr12 of DalDro are similar to that observed for Dal2, as expected based on their sequence conservation, whereas the Dro-derived residues of DalDro, Ser13-Arg16, display a different binding mode compared to the C-terminal residues of Dal2 (Fig. 5a, Supplementary Fig. 7a) and a principally different binding mode compared to Dro which binds in the orientation typical for the Type II PrAMPs (Supplementary Fig. 7b)[36]. Specifically, the sidechains of His14 and Arg16 of DalDro stack upon the nucleobases of the 23S rRNA residues A2062 and A752, respectively (Supplementary Fig. 7c). The density for Gly1 of DalDro was clearer than in the Dal2-SRC structure, showing that it is sandwiched between Lys4 of the peptide and C2573 of the 23S rRNA (Supplementary Fig. 7d). As expected from a Type I PrAMP, the N-terminus of DalDro occludes the binding site of the CCA-end of the A-site tRNA (Supplementary Fig. 7e). In addition, analogous to Dal2, we observe U2585 displacing the fMet-moiety of the P-site tRNA (Supplementary Fig. 7f), with the concomitant inversion of A76 (Supplementary Fig. 7f, g). However, unlike for Dal2, the inversion does not lead to a flipping of A2602, but rather the base of A2602 is shifted away from the ribose of A76 (Supplementary Fig. 7g, h). No density for the fMet moiety attached to the tRNA's A76 is observable,

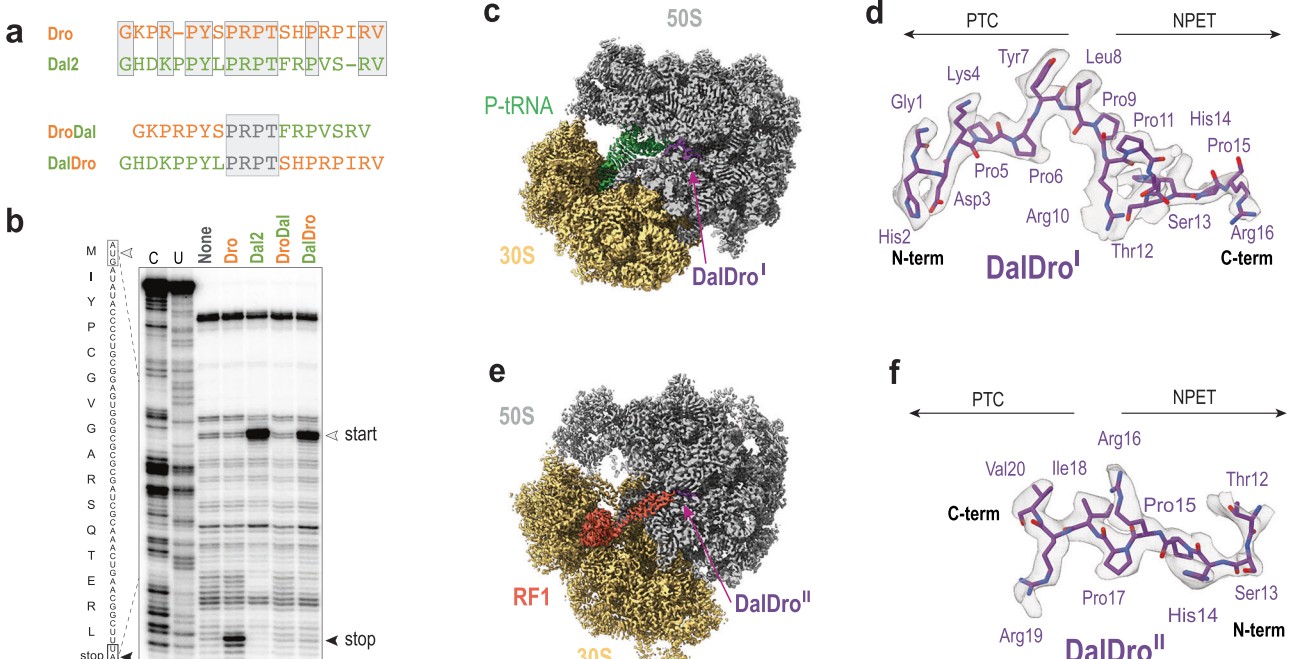

**Fig. 5 | Replacing the N-terminal segment of the Type II PrAMP Dro switches its mode of action and flips its binding orientation in the ribosome. a** Top: Sequence alignment of Type II PrAMP Dro and Type I PrAMP Dal2 (modified from Fig. 3a to emphasize their similarity). The similar/identical amino acids between the two PrAMPs are boxed. Bottom: Two chimeric peptides in which either the C-terminal sequence of Dro (DroDal) or the N-terminal sequence of Dro (DalDro) are replaced with the corresponding sequences from Dal2. **b** Toeprinting analysis of the modes of action of the native and chimeric PrAMPs. Open and closed arrowheads indicate toeprint bands corresponding to ribosomes stalled at start and stop codons, respectively. C, U- sequencing reactions. Shown gel is a representative of two independent experiments that produced converging results. Note that variation in appearance of a single or double toeprinting band (compare with Figs. 3b or 6b) is common in the toeprinting experiments. **c** Transverse section of the cryo-EM map of the DalDro-SRC (30S, yellow; 50S gray) with P-tRNA (green) and DalDro in the Type I orientation (purple) shown. **d** Cryo-EM map density (transparent gray surface) and molecular model (purple) for DalDro in Type I orientation (DalDro^I). The orientation of the PrAMP relative to the PTC and NPET is indicated by the arrows above. **e** Transverse section of the cryo-EM map of the DalDro-SRC (30S, yellow; 50S gray) in the Type II mode with RF1 (red) and DalDro in (purple) shown. **f** Cryo-EM map density and molecular model (purple) for DalDro in Type II orientation (DalDro^II).

even at low thresholds, suggesting that it is highly flexible and does not adopt a defined conformation.

Observing DalDro stalling the ribosome at the start codons and finding this peptide in a typical Type I orientation in the majority of the ribosomal particles indicates that replacing only a few N-terminal residues of the Type II Dro with those from Type I Dal2 is sufficient to switch the mode of action and binding of the PrAMP in the NPET.

Although toeprinting suggested that DalDro acts as a Type I PrAMP arresting the ribosomes at the start codon (Fig. 5b), in silico sorting of the cryo-EM data also revealed a minor population (26% of the particles) containing a P-site bound deacylated tRNA, additional density for RF1 in the A-site, as well as the DalDro peptide, which could be refined to an average resolution of 3.1 Å (Fig. 5e, Supplementary Figs. 5, 8). Strikingly, in these particles, DalDro was found bound within the NPET in the Type II orientation (Supplementary Fig. 8a, b). The PrAMP's cryo-EM density was sufficient for the unambiguous modeling of the nine C-terminal residues (Thr12-Val20) of DalDro (Fig. 5f), whereas the remaining 11 N-terminal residues protruding down the NPET (Gly1-Pro11) were poorly ordered. Within the limitations of the resolution, the conformation of the C-terminal region of DalDro was identical to that observed previously for Dro[36] (Supplementary Fig. 8b), as were the interactions between DalDro, the deacylated P-tRNA and RF1 in the A-site, including the contacts between the penultimate Arg19 of DalDro (Arg18 in Dro) and Gln235 of the GGQ motif of RF1 (Supplementary Fig. 8c). One difference between DalDro and Dro is that DalDro lacks the N-glycosylation on Thr12 that was present in the equivalent Thr residue (Thr11) of the Dro peptide utilized to obtain the Dro-SRCs structures[35]. Likely as the result of the lack

of this modification, binding of DalDro within the exit tunnel does not induce the shift in the 23S rRNA U2609 residue that breaks the base-pair with A752 as seen upon binding of the glycosylated Dro (Supplementary Fig. 8d)[36].

Although we were unable to find the reaction conditions that could demonstrate DalDro-induced arrest at the stop codon, cryo-EM analysis revealed the coexistence of two populations of the ribosome-PrAMP complexes with the peptide bound in two opposite orientations in the NPET. This finding is likely to reflect the comparable affinities of DalDro for the NPET in the Type I and Type II modes of binding.

## A single amino acid substitution inverts the PrAMP orientation in the NPET

Three sequence elements distinguish the primarily Type I DalDro from a typical Type II Dro: in DalDro two extra residues, His and Asp, are inserted after the first Gly, the Arg4 residue is missing, and the Ser preceding the PRPT motif is replaced with Leu (Fig. 6a). To examine the contributions of each of these individual elements to the PrAMP's mode of binding and action, we first tested the effects of insertions or deletions in the chimeric DalDro peptide. Simultaneously deleting His2 and Asp3 (DalDroΔHD) or even solely removing Asp3 (DalDroΔD) was sufficient to convert Type I DalDro into a Type II PrAMP (Fig. 6b). Similarly, inserting the Arg residue present in Dro into DalDro (DalDroR), yielded a PrAMP capable of arresting the ribosome at the stop codon – the hallmark of the Type II mode of action. Finally, simply replacing the DalDro-specific Leu8 residue with the Dro-specific Ser transformed the initiation arrest-mediator DalDro into the termination

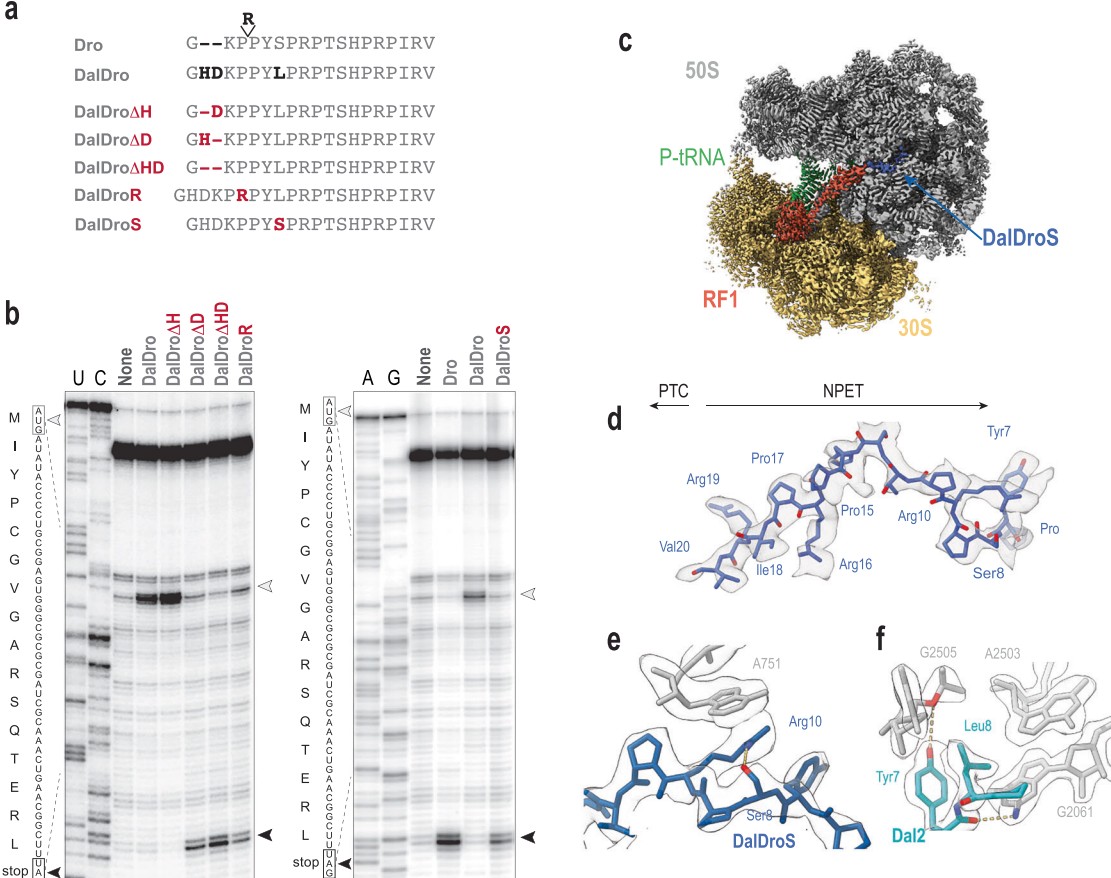

**Fig. 6 | Minimal sequence variations are sufficient for reorienting a PrAMP in the exit tunnel and transforming its mode of action. a** Sequences of Dro, the chimeric DalDro and DalDro variants. Sequence differences between Dro and DalDro are shown in bold. The introduced amino acid changes in DalDro are indicated in red. **b** Toeprinting gels showing ribosome arrest caused by the indicated PrAMPs. Toeprint bands corresponding to ribosomes arrested at the start or stop codons of the model template are indicated with open and closed arrow heads, respectively. Sequencing reactions are indicated with capital letters. Gels are representative of 3 independent experiments. **c** Transverse section of the cryo-EM map of the DalDroS-SRC (30S, yellow; 50S gray) with P-tRNA (green), RF1 (red) and DalDroS (blue) shown. **d** Cryo-EM map density (transparent gray surface) and molecular model (blue) for DalDroS. **e** Intramolecular hydrogen bond between Ser8 and Arg10 of DalDroS accommodates Arg10 into a position enabling a stacking interaction with A751 of the 23S rRNA. **f** Dal2 (light blue) Leu8 inserts into the pocket formed by 23S rRNA residues G2061, A2503 and G2505.

inhibitor DalDroS (Fig. 6b). The results of these experiments demonstrate that small changes in the peptide structure, such as deletion, insertion, or replacement of single amino acid residues, are sufficient to reverse the mode of action of a PrAMP, switching it from a translation initiation arrest peptide to an inhibitor of translation termination.

To investigate how a single substitution (Leu8Ser) can change the mechanism of action of DalDro from targeting predominantly initiation into DalDroS that targets termination, we determined a cryo-EM structure of a DalDroS-SRC (Fig. 6c). In silico sorting of the cryo-EM data revealed a single homogeneous population of ribosomes containing a P-site bound tRNA, RF1 in the A-site and DalDroS peptide bound within the exit tunnel (Supplementary Fig. 9). After the structure was refined to an average resolution of 2.8 Å (Supplementary Fig. 10), the density for DalDroS allowed for 15 residues (Pro6-Val20) to be modeled, with only the N-terminal 5 residues lacking clear density (Fig. 6d). This contrasts with the density for the DalDro peptide in the Type II pose, where only nine C-terminal residues (Thr12-Val20) were visualized (Fig. 5f), suggesting that the Leu8Ser substitution stabilizes the binding of the N-terminal region of DalDroS in the NPET. Surprisingly, Ser in DalDroS (equivalent to Ser7 in Dro[36]) does not establish interaction with the ribosome but forms an intramolecular hydrogen bond with the sidechain of Arg10 stabilizing the conformation of the

peptide that is more conducive to the Type II binding (Fig. 6e). In the Type I pose of DalDro, the sidechain of Leu8 stacks on Tyr7 and inserts into a pocket formed by G2061, A2503 and G2505 (Fig. 6f), whereas the sidechain of Ser of DalDroS is too short and polar to form stable hydrophobic and van der Waals interactions within this pocket.

Altogether, the results of the structural analysis of the ribosome bound by the two PrAMPs differing by a single amino acid residue (DalDro and DalDroS) show that a minimal change in the peptide sequence is sufficient to dramatically switch its mode of binding, flipping its orientation in the ribosomal NPET, creating a principally different sets of contacts with the ribosome components and, as a result, inhibiting different steps of translation.

## Discussion

Previous structural and biochemical analyses uncovered two contrasting mechanisms employed by Type I and Type II PrAMPs to engage and inhibit the ribosome. In this work we show that the structural boundaries separating the two types can be so narrow such that even a single amino acid change allows a PrAMP to flip its orientation in the NPET and acquire an alternative mode of binding to the ribosome and manner to interfere with translation.

All the proteins synthesized by the ribosome traverse the NPET in the universal N-terminus-first orientation defined by the chemical logic

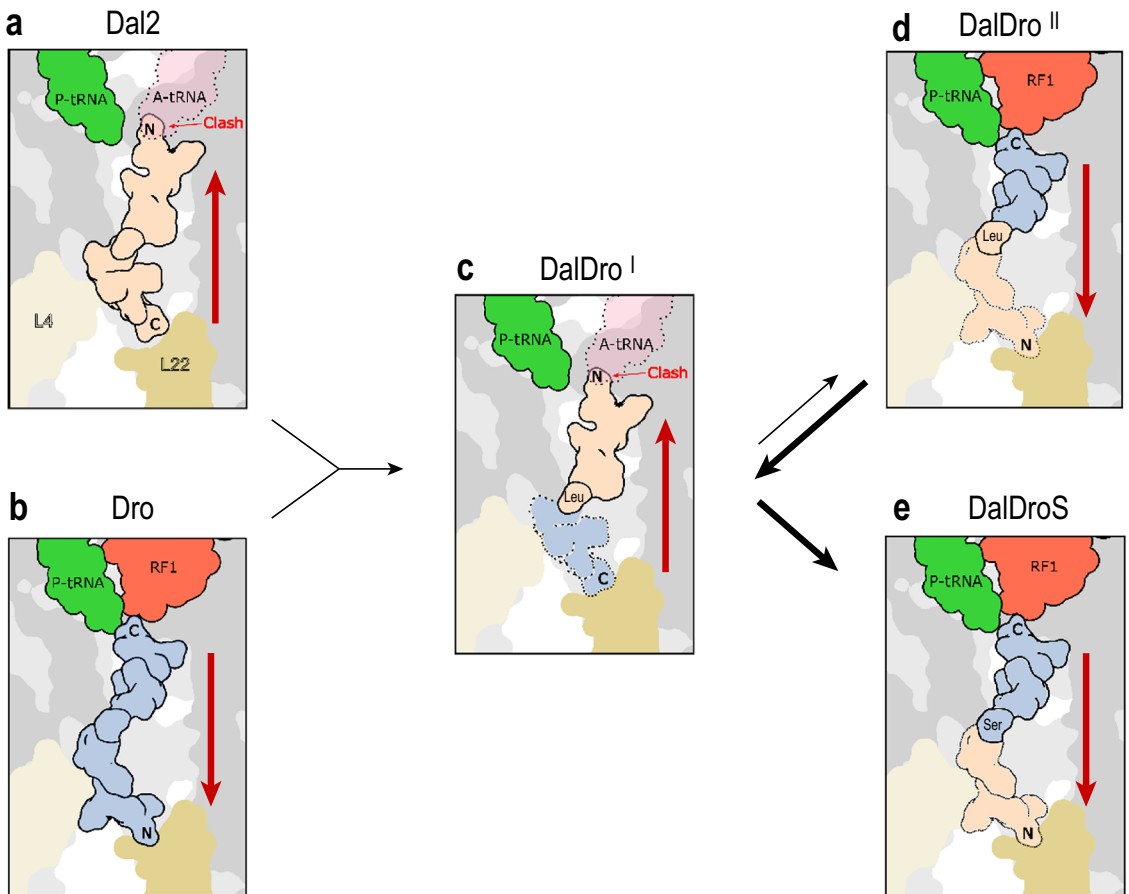

**Fig. 7 | Model for conformational flipping of PrAMPs within the ribosomal tunnel.** Schematic representation of (**a**) Type I Dal2 bound within the tunnel such that accommodation of the A-site tRNA is prevented due to a steric clash with the N-terminus of Dal2, **b** Type II Dro bound within the tunnel such that termination is inhibited by trapping RF1 on the ribosome, **c** The hybrid PrAMP DalDro binds predominantly as a (**c**) type I PrAMP, however, **d** a flipped conformation where the Type II orientation is also observed for DalDro. **e** A single point mutation of Leu in DalDro to Ser to generate DalDroS precludes binding as Type I PrAMP favoring the binding of DalDroS within the tunnel exclusively with a Type II orientation to interfere with translation termination.

of protein synthesis. However, posttranslational interactions of polypeptides with the NPET may take place with opposing orientations, depending on whether it is the N- or the C-terminus of the polypeptide that is threaded through the tunnel towards the PTC[9–16]. Posttranslational binding in the NPET requires that the respective segment of the polypeptide adopts a predominantly extended conformation conducive to its fitting within the tunnel's narrow aperture. The tight binding of the extended polypeptide chain in the NPET relies on interactions of the amino acid side chains with rRNA and ribosomal protein residues that line the tunnel walls. The general structural properties of PrAMPs make them particularly well-suited for binding within the NPET. Structurally-constrained proline residues are scattered through the length of the PrAMPs, discouraging formation of stable secondary structure elements and instead favoring unfolded linear conformations[51,52]. In turn, the side chains of the abundant arginine residues can establish electrostatic, stacking and H-bonding interactions with the sugar-phosphate backbone and nitrogen bases of the rRNA nucleotides in the NPET[35,38,41,42]. The isotropy of these physicochemical characteristics imposed upon the PrAMPs by their abundant proline and arginine residues should allow the peptides to enter the NPET in either orientation and form relatively stable complexes irrespective of whether the N- or the C-terminus of the peptide initiates the entrance to the tunnel. Our studies of the Dro-like peptides show that for some PrAMPs, the binding energy in the opposing orientations may be close to equilibrium so that the predominance of one of the alternative binding poses is defined by subtle differences in the peptide's structure. Similar considerations may account for the reported binding of sequence-similar PrAMPs in opposite orientations to the DnaK chaperone, the formerly presumed target of these peptides in bacteria[53].

The inverted binding of Type I PrAMPs, such as Dal2, to the ribosome relies heavily on multiple interactions of the peptide's N-terminal segment with the PTC A site where it blocks accommodation of the A-site tRNA (Fig. 7a). This could explain why substituting the N-terminus of the Type II Dro (Fig. 7b) with that of the Type I Dal2, shifts the equilibrium between the two alternative modes of binding of the PrAMP in the NPET in favor of Type I orientation, as observed for the predominant pose of the chimeric DalDro (Fig. 7c). Yet, the observation of the Type II mode of DalDro binding in a subpopulation of ribosomes vividly illustrates the similarity of binding energies of the PrAMP in two alternative orientations (Fig. 7). As the result, a minimal change in the structure of a PrAMP, such as a single amino acid addition, deletion or substitution could be sufficient to flip the peptide's orientation in the NPET resulting in a dramatic change in its mode of action (Fig. 7e).

Residue Thr11, located in the conserved PRPT motif, is posttranslationally glycosylated in the natural Dro[54]. A similar modification is also found in some Type I PrAMPs[55]. Although Thr glycosylation is not essential for Dro's activity, the sugar residues establish specific interactions with the NPET[36]. It is unclear whether any of the newly

identified Dro-like PrAMP are posttranslationally modified but it is conceivable that the modification could be an additional factor influencing the equilibrium between the two binding modes.

The narrow sequence boundaries between Type I and Type II PrAMPs open attractive evolutionary opportunities for the host. As translation-inhibiting antibiotics, the termination-arresting Type II PrAMPs offer a general advantage of not only arresting the peptide-bound ribosome at the stop codon of an ORF, but also taking offline all the trailing ribosomes on the same mRNA[43]. However, the action of Type II PrAMPs depends on the properties of the RFs that could be easily mutated to render bacteria resistant to the PrAMP[35,56]. In contrast, the action of Type I PrAMPs is RF-independent, and resistance cannot be easily achieved by target site mutations[34]. If the host is infected by a new or evolved pathogen that is impervious to the action of Type II PrAMPs, it is only a few or even one mutation away from converting the Type II PrAMP into a Type I whose action would not be affected by the properties of the pathogen's RFs. Noteworthy, insects often encode multiple isoforms of PrAMPs that differ in just a few amino acids. It might be a strategy for generating a repertoire of PrAMPs, with some capable of flipping, if the need should arise. Thus, not only the small size of the PrAMPs, but also the ability to switch their mode of binding and action by a minimal number of mutations could benefit the host in the ever-going race with the evolving bacterial pathogens.

## Methods

### Materials and reagents
All peptides were synthesized by NovoPro Biosciences Inc., majority of them have purity >90% (Supplementary Table 3). Oligonucleotides were from Integrated DNA Technology. All the chemicals and enzymes were from Thermo Fisher Scientific or New England Biolabs.

### Identifying Dro homologs by mining protein databases
For the initial searching, the peptide sequence GKPRPYSPRPTSHPR-PIRV of Drosocin (Dro) from *Drosophila melanogaster*[54] was used as a query with the BLASTP algorithm of the NCBI BLAST suite, to search the non-redundant protein sequences database. The tBLASTN search was also performed with Dro peptide sequence against fly Genome Assembly with default setting in Flybase (https://flybase.org/). The search was then expanded using the newly found distinct Dro-like PrAMPs as probes. After manual curation, the hits corresponding to the predicted pre-pro-proteins encoding single or multiple Dro-like PrAMP were retrieved (Supplementary Table 1). The boundaries of the predicted Dro-like and other co-encoded PrAMPs were deduced based on the previously identified furin-like processing signals and the Dro-like PrAMP sequences retrieved from literature review[45]. To identify unique PrAMPs, non-redundant PrAMP sequences were used for multiple sequence alignment of PrAMPs with the MUSCLE algorithm in Geneious Prime software (version 2023.2.1)[57]. The alignments were additionally manually adjusted.

Unique PrAMP-like isoforms were used for phylogenetic analysis, the aligned PrAMP isoform sequences were further trimmed to eliminate ambiguous regions before IQTree building (http://iqtree.cibiv.univie.ac.at/) based on maximum likelihood method. Type I PrAMPs Oncocin, and Pyrrhocoricin were also included in tree generation, the proline-arginine rich cathelicidin PR-39 from pig were included as an outgroup. iTOL (https://itol.embl.de/) was used for tree visualization and editing.

### In vitro translation and toeprinting analysis
The sfGFP reporter protein was expressed in the *E. coli* cell-free transcription-translation system based on cell lysate (New England Biolabs, Cat No. E5360S). The DNA template was generated by PCR-amplifying the T7 promoter-sfGFP gene segment of the pY71sfGFP plasmid[58] using forward primer T7 (TAATACGACTCACTATAGGG) and reverse primer

sfGFP-UAG-R (GCCGGTCGACCTATTTTTCGAACTG) as previously described[37]. Transcription-translation reactions containing 200 ng of the DNA template were carried in a final volume of 5 µl supplemented with 0.5 µl of 500 µM PrAMPs (dissolved in $H_2O$) or 0.5 µl of $H_2O$ in the control samples. Fluorescence of sfGFP was recorded every 10 min in a 96-well plate reader (Tecan) at 37 °C for 2 h ($\lambda$ exc = 485 nm, $\lambda$ em = 535 nm). The 50 min time point, corresponding to the end of the slope stage of the reaction, was used to calculate the extent of inhibition of translation by PrAMPs.

Toeprinting experiments were carried out in the PURExpress transcription-translation system (New England Biolabs, Cat No. E6800L) following the procedure described previously using the *yrbA*-fs15 DNA template with the sequence TAATACGACTCACTATAG GGCTTAAGTATAAGGAGGAAAACAT**ATG**ATATACCCCTGCGGAGTGG GCGCGCGATCGCAAACTGAACGGCTT**TAG**GCCGACCTCGACAGTTGG ATTCACGTGCTGAATCCTGATGCGATGTCGAGTTAATAAGCAAAATT-CATTATAACC[37]. PrAMPs were added from 500 µM stock solutions to a final concentration of 50 µM.

### Preparation of complexes for structural analysis
The PrAMP stalled-ribosomal complexes were generated by in vitro translation reactions in the PURExpress In vitro Protein Synthesis Kit (NEB) as described by the manufacturer. Complex formation reactions were carried out on MLIF mRNA template (UAAUACGACUCA-CUAUAGGGAGACUUAAGUAUAAGGAGGAAAAAAU**AUG**AUAUUCUU-G**UAA**AUGCGUAAUGUAGAUAAAACAUCUACUAUUUAAGUGAUAGA AUUCUAUCGUUAAUAAGCAAAAUUCAUUAU) in a 75 µl of reaction in presence of 50 µM of the respective peptide. The reaction was incubated for 15 min at 37 °C. Ribosome complexes were isolated by centrifugation in 900 µl of sucrose gradient buffer (containing 40% sucrose, 50 mM HEPES-KOH, pH 7.4, 100 mM KOAc, 25 mM $Mg(OAc)_2$ and 6 mM 2-mercaptoethanol) for 3 h at 4 °C with 80,000 × $g$ in a Optima Max-XP Tabletop Ultracentrifuge with a TLA 120.2 rotor. The pelleted complex was resuspended in Hico buffer (50 mM HEPES-KOH, pH 7.4, 100 mM KOAc, 25 mM $Mg(OAc)_2$) supplemented with 50 µM of the respective peptide, then incubated for 10 min at 37 °C, similarly to that described previously[36,59].

### Preparation of cryo-EM grids
Cryo-EM grids were prepared by applying 3.5 µl of the PrAMP-stalled ribosomal complexes onto freshly glow-discharged Quantifoil R3.5/1 grids (copper, 300 mesh, with an additional 3 nm carbon layer; Product: C3-C19nCu30-01). The glow discharge was performed using a GloQube® Plus system (Quorum Technologies) at 25 mA for 30 s, in a negatively charged atmosphere. Vitrification of the samples was carried out with a 1:2 ethane-to-propane mixture using a Vitrobot Mark IV (Thermo Scientific). The chamber was maintained at 100% relative humidity and 4 °C. Blotting was performed for 3.5 s at blot force 0, using Whatman 597 filter paper. After vitrification, the grids were loaded into autogrid cartridges and stored in liquid nitrogen until further use.

### Data acquisition
Data acquisition was conducted on a Titan Krios G3i transmission electron microscope (Thermo Fisher Scientific/FEI) operating at the Center for Structural Systems Biology (CSSB), Hamburg. The microscope was operated in Fringe-Free Imaging (FFI) mode, equipped with a K3 direct electron detector and a BioQuantum energy filter with a 20 eV slit width. Prior to data collection, gain reference and GIF fine-centering were completed. Automated data acquisition was carried out using EPU software (v3.2.0.4775REL). Movies were captured at a nominal magnification of 105,000×, corresponding to a calibrated pixel size of 0.832 Å (0.416 Å in super-resolution mode, binned 2× via EPU). The dataset was collected using defocus values ranging from −0.3 µm to −1.0 µm in 0.1 µm increments between holes. Each exposure

lasted 1.95 s in nanoprobe mode, during which 35 frames were recorded at a dose rate of -1.14 electrons per frame per Å$^2$, resulting in a total accumulated dose of approximately 40 electrons per Å$^2$ (-15 e$^-$/px/s over vacuum). A 70 μm C2 aperture and beam spot size 7 were used. Objective lens astigmatism was corrected to below 1 nm, and coma-free alignment was refined to under 50 nm using Sherpa's AutoCTF module (v2.11.1). In total, 12,485 gain-corrected TIFF micrographs were acquired for the Dal2-70S complex, 7819 for the DalDroS-70S complex and 16,758 for the DalDro-70S complex.

## Cryo-EM data processing

RELION v5.0.0[60,61] was used for image processing, unless otherwise specified. For motion correction, RELION's implementation of MotionCor2 with 7 × 5 patches[62], and for initial contrast transfer function (CTF) estimation, CTFFIND version 4.1.14[63], were employed. Particle picking was performed using crYOLO[63] and the particle coordinates were then imported into RELION. After 2D classification, all ribosome like particles were selected, extracted and 60 Å low pass filtered 70S ribosome (PDB ID 7K00)[64] was used as reference to perform 3D consensus refinement of these particles. With this 3D refined map, 3D classification was performed without angular sampling. All classes that contained 70S ribosomes at high resolution were used for further processing. Particles with homogenous 3D class distribution were re-extracted using smaller pixel size and subjected to 3D refinements. Subsequently, CTF refinements were performed to correct for anisotropic magnification, defocus and astigmatism, beam tilt, trefoil and higher order aberration followed by Bayesian polishing if not stated differently[65]. For partial signal subtraction, masks around the region of interest were created. Masking of 3D maps was done using soft mask to avoid artificial correlation and extended to several pixels to avoid overlap with volume. Sorting schemes for the respective datasets are shown in Supplementary Figs. 1, 5, and 9.

## Generation of molecular models

The molecular models were based on the *E. coli* 70S ribosome (PDB ID 7K00)[64]. Starting models with individual chains of ribosomal proteins and rRNA were rigid body fitted using ChimeraX[66] and modeled using Coot 0.9.8.96[67,68] from the CCP4 software suite version 9.0[69]. Model refinement was done using Servalcat[70]. The final molecular models were validated using Phenix comprehensive cryo-EM validation tool in Phenix 1.20–4487[71].

## Figure preparation for cryo-EM data

Particle orientations and their distribution was determined and plotted using Relion v5.0.0[60,61]. The Molprobity server[72] was used to calculate map vs model cross correlation at Fourier Shell Correlation (FSC$_{0.5}$) for all maps (Supplementary Figs. 2, 6, and 10). UCSF ChimeraX v1.8[66] was used to isolate densities, color zone maps and visualize density images. Models were aligned using PyMol version 3.0 (Schrödinger). Figures were assembled using Inkscape v1.3.

## Reporting summary

Further information on research design is available in the Nature Portfolio Reporting Summary linked to this article.

## Data availability

Cryo-electron microscopy maps have been deposited at the Electron Microscopy Data Bank (EMDB) under accession codes EMD-55845 (Dal2-SRC), EMD-55844 (DalDro$^i$-SRC), EMD-55847 (DalDro$^t$-SRC) and EMD-55849 (DalDroS-SRC) with the associated molecular models deposited at the Protein Data Bank (PDB) with accession numbers 9TEX, 9TEW, 9TEY and 9TEZ, respectively. Source data are provided with this paper as a Source Data file.

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

## Acknowledgements

We thank Gemma Atkinson (Lund University) and Ariya Chang (Whitney M. Young High School) for help at early stages of this project. This work was supported by a grant R01 AI162961 from the National Institutes of Health (to A.S.M. and N.V.-L.), and by the Deutsche Forschungsgemeinschaft (DFG, German Research Foundation) WI3285/12-1 (to D.N.W.). Part of this work was performed at the Multi-User CryoEM Facility at the Centre for Structural Systems Biology, Hamburg, supported by the Universität Hamburg and DFG grant numbers (INST 152/772-1|152/774-1|152/775-1|152/776-1|152/777-1 FUGG), the Federal Ministry of Education and Research (BMBF) and the DLR Projektträger (project SEEK 01KX2220). We acknowledge financial support from the Open Access Publication Fund of Universität Hamburg.

## Author contributions

W.H. carried out genome mining and bioinformatics analyses, performed toeprinting experiments, interpreted the data and contributed to writing the paper. M.J.B. generated the samples for structural analysis, performed the cryo-EM analysis, as well as generated and refined the molecular models, with help from H.P. H.A.S. prepared cryo-EM grids, screened and collected the cryo-EM data. D.K. carried out toeprinting experiments. C.B. carried out genome mining and bioinformatics analyses and contributed to writing the paper. W.H., D.N.W., N.V.-L., and A.S.M. wrote the manuscript. N.V.-L. and A.S.M. conceived the project. N.V.-L., A.S.M., and D.N.W. supervised the project.

## Funding

## Competing interests

The authors declare no competing interests.
