## [Transparent Peer Review File · Nature Communications]

Flipping antimicrobial peptides in the exit tunnel of the bacterial ribosome

Corresponding Author: Professor Daniel Wilson

Version 0:

Reviewer comments:

Reviewer #1

(Remarks to the Author)

In this interesting study, Huang et al. investigate the mechanisms by which proline-rich antimicrobial peptides (PrAMPs) inhibit translation. Building on their recent work, the authors identify and characterize previously unstudied Type II Drosocin-like PrAMP homologs from related organisms. By mining available genome databases, they identify several candidate peptides in *Drosophila* and validate their activity experimentally. Surprisingly, despite their overall sequence similarity to canonical Type II peptides, toeprinting assays and cryo-EM analyses reveal that these peptides inhibit translation at the initiation step rather than at termination. Moreover, minimal alterations in the sequence of these peptides were sufficient to flip their orientation within the ribosomal tunnel and switch between the two stalling modes. Together, the biochemical and structural data are of high quality and provide strong support for the proposed model, making this work suitable for publication in *Nature Communications*. There are only minor comments and questions to be addressed prior to publication:

1. In several places, the manuscript refers to Type II PrAMPs as “termination inhibitors.” However, the structures appear to capture post-nascent-chain release states. Would it be more accurate to describe these peptides as inhibitors of ribosome recycling rather than termination per se?
2. Type II PrAMPs trap ribosomes with release factors within a relatively narrow window after peptidyl-tRNA hydrolysis and nascent-chain release. A likely limiting step is the diffusion of the nascent chain out of the peptide exit tunnel. Given that peptide release has occurred, why does RF2 remain bound to the ribosome?
3. In Fig. 1, the A- and P-site labels are difficult to see. Adding a thicker outline or increasing contrast would improve readability.
4. In several toeprinting assays, multiple bands are observed at the termination step (e.g., Fig. 4b), whereas other assays show a single predominant band (e.g., Fig. 5b). Do these differences reflect two distinct stalled ribosomal states?
5. The DalDro_termination model file appears to lack mRNA, despite visible density in the 30S subunit.
6. The C-terminal helix of L31 could potentially be trimmed in the model associated with the 50S-focused map, as it appears to be associated with the head of the small subunit.
7. Fig. 5b does not have a sequencing ladder map/key.
8. The supplementary information file contains only one table, yet the text refers to Tables S1–S6. Are several supplementary tables missing?

For example:

The text references “Table S1” in the line “In total, we found 26 unique Dro-like PrAMPs...,” but Table S1 appears to correspond to EM validation statistics. Check that the PrAMP list table and EM validation tables are correctly labeled and referenced.

The Methods reference “Supplementary Table 6” in the “Preparation of complexes for structural analysis” section. There appears to be an in-text citation formatting issue in the first line of the “Figure preparation” section of the Methods.

Reviewer #2

(Remarks to the Author)

I co-reviewed this manuscript with one of the reviewers who provided the listed reports. This is part of the *Nature Communications* initiative to facilitate training in peer review and to provide appropriate recognition for Early Career Researchers who co-review manuscripts.

Reviewer #3

(Remarks to the Author)

Insects and mammals inhibit bacterial pathogens by the expression of small peptides that bind to the nascent chain exit channel of the ribosome and prevent either translation initiation or termination. These PrAMPs can bind with either their N terminus towards the exit of the tunnel (Type II) or opposite with the N terminus facing the peptidyl transferase center (Type I; inverted). These PrAMPs are rich in prolines and arginines and are hard to distinguish based alone on sequences. In this manuscript the authors mined insect genomes for sequences similar to the Type II drosocin (Dro) PrAMP and find that although this *Drosophila* PrAMP has characteristics of a Type II, they find it functions like a Type I and binds in an inverted orientation on the ribosome and inhibits initiation. The authors used bioinformatics to identify new sequences in insects and then test these peptides using biochemistry and structural biology. Overall, this is an interesting story.

The authors use biochemical and reporter assays in *E. coli* to test 2 PrAMPs from *Sophophora* and 7 PrAMPs from *Drosophila*. The results from these assays are not entirely consistent. While the toeprint assays unequivocally show that *Sophophora* PrAMPs inhibit at translation initiation and *Drosophila* PrAMPs inhibit at termination (Fig 2b), the reporter assays show variable levels of inhibition at each of these stages. For example, Dro only shows ~60% inhibition at initiation while levels of ~10-35% inhibition is seen for *Sophophora* PrAMPs. What is the reason for why almost complete inhibition does not occur for the *Sophophora* PrAMPs? Are the peptides not getting into cells or are they recognized and degraded by proteases? There could be a reason for why these results are not entirely consistent but the authors do not mention any potential reasons and further, state these results are consistent. Prior studies by the Wilson lab showed that the O-glycosylation of Thr11 of Dro is needed for cellular uptake and to bind efficiently to the ribosome (ref 36). Could this be the reason? And if so, why not include the glycosylation for this study? Also, the authors look at inhibition of activity (toeprint) and reporter assays to report on "activity" of these peptides. But what about affinity? Could the differences simply be because of altered affinities of these different peptides (and the chimeras discussed later on in the paper) to the ribosome and that activity can be restored if one adds the amount of peptide that is well above their K_d ? It seems like this is a missing part of the story especially in light of the potentially conflicting data.

The authors mention that in prior studies of Dro bound to ribosome, they show two populations of Dro where they differed in the orientation of Dro (the authors reference 36). To the best of my knowledge, this is not stated in that publication so this is surprising. It does raise concern that if Type II peptides can adopt both positions, perhaps this means that it is not so surprising that they find Dal2 is in an inverted position? This point is confusing.

The authors next test chimeric DalDro and DroDal peptides to determine which part of the N or C terminus are important for activity. While DroDal is no longer active, DalDro changes from a Type II to a Type I based upon toeprint analysis. Since the authors used a reporter assay for prior studies, it makes sense to also test these chimeras in this assay.

The interactions between DalDroS and 23S rRNA are shown in Fig 5e and 5f. Maps should be shown for both (I believe surfaces are shown in Fig 5f).

Fig 6 shows a model for how these peptides engage the NPET and L4 and L22 r-proteins are shown but are never discussed in this manuscript despite the fact that the C-terminus of Dal2 seems to clearly interact with L4 and L22 (Fig 6a). This needs to be discussed.

The last sentence of the Intro is intriguing in that the authors imply that flipping the orientation of the PrAMP could be a strategy by insects to combat bacterial pathogens. This is never expanded on anywhere else in the manuscript. Is this known? Or would insects just encode both Type I and Type II PrAMPs and not rely upon costly mutations?

Minor-p12 typo (whreas)

Reviewer #4

(Remarks to the Author)

This is a very comprehensive study demonstrating how a small number of mutations can alter—or even flip—the binding mode of PrAMPs within the bacterial ribosome. There are a few questions I would like the authors to address.

- The work focuses on Drosocin variants in *E. coli*. Are there mechanistic insights that could guide rational mutant design to modulate or flip binding modes across different bacterial species and across distinct PrAMP families, which may exhibit sequence and structural diversity?
- Additionally, do the reported mutations influence membrane uptake efficiency? Since intracellular transport (e.g., via SbmA) is essential for PrAMP activity, it would be important to clarify whether the mutations affect ribosome binding exclusively or also alter cellular entry, thereby contributing to differences in antimicrobial activity.
- Can the equilibrium between flipped orientations (e.g., observed for DalDro) be characterized quantitatively to support the proposed near-equivalent binding energies (e.g., K_D values)?

Version 1:

Reviewer comments:

Reviewer #1

(Remarks to the Author)

The authors have adequately addressed our comments and we recommend the publication of this manuscript.

Reviewer #2

(Remarks to the Author)

Reviewer #4

(Remarks to the Author)

Thank you to the authors for their efforts in addressing my questions and clarifying the limitations of the experimental methods for further mechanistic studies. I recommend publishing the manuscript.

Arbitrating Referee

In my view, reviewer 3 has not raised any substantive technical concerns. The points raised appear could reflect a misunderstanding or misinterpretation of the footprinting results rather than underlying methodological issues.

The authors have clearly addressed these points in their response and revisions, and I believe the concerns raised by reviewer 3 have been adequately resolved.

REVIEWER COMMENTS

Reviewer #1:

In this interesting study, Huang et al. investigate the mechanisms by which proline-rich antimicrobial peptides (PrAMPs) inhibit translation. Building on their recent work, the authors identify and characterize previously unstudied Type II Drosocin-like PrAMP homologs from related organisms. By mining available genome databases, they identify several candidate peptides in *Drosophila* and validate their activity experimentally. Surprisingly, despite their overall sequence similarity to canonical Type II peptides, toeprinting assays and cryo-EM analyses reveal that these peptides inhibit translation at the initiation step rather than at termination. Moreover, minimal alterations in the sequence of these peptides were sufficient to flip their orientation within the ribosomal tunnel and switch between the two stalling modes. Together, the biochemical and structural data are of high quality and provide strong support for the proposed model, making this work suitable for publication in *Nature Communications*.

We are glad the reviewer found our findings interesting, and we appreciate the favorable evaluation of our experimental work.

There are only minor comments and questions to be addressed prior to publication:

1. In several places, the manuscript refers to Type II PrAMPs as “termination inhibitors.” However, the structures appear to capture post–nascent-chain release states. Would it be more accurate to describe these peptides as inhibitors of ribosome recycling rather than termination per se?

The reviewer is correct that Type II PrAMPs bind to the post-release ribosome which rises a legitimate question of whether they could be called *bona fide* inhibitors of translation termination. Nevertheless, we believe it would be inaccurate to call them “inhibitors of ribosome recycling”. Firstly, as we showed in our previous work (PMID 28741611, 33031031), sequestering RFs on a fraction of the cellular ribosomes leads to a rapid RF depletion in the cells treated with Type II PrAMPs. Because of the excess of ribosomes over RFs, the majority of the ribosomes in the cell are then stalled at the stop codons in a pre-release step, unable to hydrolyze peptidyl-tRNA thereby being incapable of terminating translation. Secondly, formally speaking, the termination step of translation is completed only after RFs dissociate from the ribosome making the ribosome available for binding of the ribosome recycling factor. Because Type II PrAMPs prevent class I RF dissociation, we believe they could be viewed as inhibitors of translation termination. For these two reasons, we want to keep referring to Type II RFs as “termination inhibitors”. However, in order to make this point less ambivalent, we revised the corresponding section of the Introduction to read as follows: “*In stark contrast to the action of Type I PrAMPs, Type II PrAMPs arrest the ribosome not at the start codon but at the stop codon. They enter the NPET after the nascent protein is released, diffuse towards the PTC, and trap the ribosome-associated class I release factors (RF1 or RF2) due to the interaction of the PrAMP’s*

penultimate arginine residue with the glutamine of the conserved GGQ motif of class I RFs. As a result, Type II PrAMPs trap RF1/RF2 on the post-release ribosome preventing their dissociation needed for the completion of the termination stage of translation {Florin, 2017 #8486; Graf, 2018 #8800; Mangano, 2020 #9445; Koller, 2023 #10167; Mangano, 2023 #10169}. Furthermore, because in bacteria ribosomes are present in excess over RFs, depletion of RFs in the Type II PrAMP-treated cells result in stalling of the majority of the cellular ribosomes at the stop codons in a pre-release stage, revealing these PrAMPs as bona fide termination inhibitors.”

2. Type II PrAMPs trap ribosomes with release factors within a relatively narrow window after peptidyl-tRNA hydrolysis and nascent-chain release. A likely limiting step is the diffusion of the nascent chain out of the peptide exit tunnel. Given that peptide release has occurred, why does RF2 remain bound to the ribosome?

The reviewer is correct in pointing out the importance of kinetics of RF dissociation in the mode of action of Type II PrAMPs. Indeed, in our understanding, the time window that allows Type II PrAMPs to trap class I RFs on the post-release ribosome is defined by the kinetics of the nascent chain extraction from the exit tunnel, the rate of the PrAMP diffusion up the tunnel and the residence time of the RF1/RF2 on the post-release ribosome. If RF1/2 remains bound to the post-release ribosome long enough, it would be trapped by the PrAMP. However, if RF1/2 dissociates sooner than the C-terminal tail of the Type II PrAMP reaches the PTC, the PrAMP will fail to trap RFs on the ribosome. Currently, neither the parameters of nascent chain extraction nor those of PrAMP diffusion through the tunnel are known. Nevertheless, dissociation of RF is apparently sufficiently slow to afford Type II PrAMPs an ample opportunity to trap them on the ribosome in vivo and in vitro. Because this aspect of Type II PrAMP is fairly unclear, we chose to not discuss it in our paper.

3. In Fig. 1, the A- and P-site labels are difficult to see. Adding a thicker outline or increasing contrast would improve readability.

We adjusted the contrast of the labels according to the reviewer’s suggestion.

4. In several toeprinting assays, multiple bands are observed at the termination step (e.g., Fig. 4b), whereas other assays show a single predominant band (e.g., Fig. 5b). Do these differences reflect two distinct stalled ribosomal states?

We praise the reviewer for attention to the detail! In our many years of using toeprinting for various projects we often observe appearance of a single or a double toeprinting band on the sequencing gel which could vary from batch to batch of the PURExpress cell-free system for undetermined reasons. Because these minor variations are not critical for our results, we prefer not to discuss them in our paper. However, following the reviewer’s question, we added a

following note to the Figure 4b legend: "Note that variation in appearance of a single or double toeprinting band (compare with Figs 2b or 5b) is common in toeprinting experiments."

5. The DalDro_termination model file appears to lack mRNA, despite visible density in the 30S subunit.

Following the reviewer's comment, we have added a model for the parts of the mRNA that have visible density in the 30S subunit, namely, for L31 residues 55-66 were removed and the mRNA in the A-, P- and E-sites (5' AUA-UUC-UAA-3', with UUC being the P-site codon).

6. The C-terminal helix of L31 could potentially be trimmed in the model associated with the 50S-focused map, as it appears to be associated with the head of the small subunit.

Following the reviewer's suggestion, the C-terminal helix of L31 has been trimmed, specifically removing residues 55-66 of L31.

7. Fig. 5b does not have a sequencing ladder map/key.

Sequencing ladders have been added to the figure.

8. The supplementary information file contains only one table, yet the text refers to Tables S1–S6. Are several supplementary tables missing?

For example:

The text references "Table S1" in the line "In total, we found 26 unique Dro-like PrAMPs..." but Table S1 appears to correspond to EM validation statistics. Check that the PrAMP list table and EM validation tables are correctly labeled and referenced.

The Methods reference "Supplementary Table 6" in the "Preparation of complexes for structural analysis" section.

The missing tables have been added. All the tables have been now placed in the Supplementary Information file.

There appears to be an in-text citation formatting issue in the first line of the "Figure preparation" section of the Methods.

The formatting issue has been fixed

Reviewer #2 (Remarks to the Author):

We thank Reviewer 2 for co-reviewing our manuscript and providing feedback and advice.

Reviewer #3 (Remarks to the Author):

Insects and mammals inhibit bacterial pathogens by the expression of small peptides that bind to the nascent chain exit channel of the ribosome and prevent either translation initiation or termination. These PrAMPs can bind with either their N terminus towards the exit of the tunnel (Type II) or opposite with the N terminus facing the peptidyl transferase center (Type I; inverted). These PrAMPs are rich in prolines and arginines and are hard to distinguish based alone on sequences. In this manuscript the authors mined insect genomes for sequences similar to the Type II drosocin (Dro) PrAMP and find that although this *Drosophila* PrAMP has characteristics of a Type II, they find it functions like a Type I and binds in an inverted orientation on the ribosome and inhibits initiation. The authors used bioinformatics to identify new sequences in insects and then test these peptides using biochemistry and structural biology. Overall, this is an interesting story.

We thank the reviewer for the detailed and accurate summary of our findings and for the positive assessment of the work.

The authors use biochemical and reporter assays in *E. coli* to test 2 PrAMPs from *Sophophora* and 7 PrAMPs from *Drosophila*. The results from these assays are not entirely consistent. While the toeprint assays unequivocally show that *Sophophora* PrAMPs inhibit at translation initiation and *Drosophila* PrAMPs inhibit at termination (Fig 2b), the reporter assays show variable levels of inhibition at each of these stages. For example, Dro only shows ~60% inhibition at initiation while levels of ~10-35% inhibition is seen for *Sophophora* PrAMPs. What is the reason for why almost complete inhibition does not occur for the *Sophophora* PrAMPs? Are the peptides not getting into cells or are they recognized and degraded by proteases? There could be a reason for why these results are not entirely consistent but the authors do not mention any potential reasons and further, state these results are consistent.

The biochemical assay presented in Figure 2c was specifically designed to highlight the distinct mechanisms of action of Type I and Type II PrAMPs. Type II PrAMPs readily inhibit cellular translation but are notably poor inhibitors in cell-free translation systems. This difference arises because in the cell-free system, Type II PrAMPs bind to the ribosome only after completion of the first round of translation, when the nascent polypeptide has been released by the release factor (RF). Following this event, the PrAMP traps the RF on the ribosome. Consequently, the

first round of translation produces the reporter protein and inhibition of in vitro translation by Type II PrAMPs occurs only after the RFs, initially present in the reaction, are depleted.

In contrast, Type I PrAMPs target the initiating ribosome and therefore more effectively inhibit reporter expression in vitro, being able to bind to the ribosome before translation of the mRNA starts. Thus, the results shown in Figure 2c are fully consistent with the distinct mechanisms of action of the two PrAMP classes.

In the original submission, we noted that “Type II PrAMPs are notably poor inhibitors of in vitro protein synthesis because they bind to the terminating ribosome only after the completion of the first round of translation {Krizsan, 2014 #8236;Koller, 2023 #10167;Mangano, 2023 #10169}.” However, to avoid potential confusion, we have expanded this explanation. The revised text now reads: “By arresting the initiating ribosome, Type I PrAMPs are known to readily interfere with the in vitro translation of a reporter protein. In contrast, Type II PrAMPs are notably poor inhibitors of in vitro protein synthesis because they bind to the terminating ribosome only after the completion of the first round of translation and exert their inhibitory action only after RFs are exhausted in a cell-free translation reaction”.

Prior studies by the Wilson lab showed that the O-glycosylation of Thr11 of Dro is needed for cellular uptake and to bind efficiently to the ribosome (ref 36). Could this be the reason? And if so, why not include the glycosylation for this study?

The reviewer is correct that glycosylation may influence both the uptake of Dro and its binding to the ribosome. However, we were unable to include glycosylated Dro in this study because its chemical synthesis is extremely challenging. The previous study cited by the reviewer (ref. 36) was performed using the final batch of glycosylated Dro synthesized in the laboratory of Prof. Kaur at the National Institute of Immunology (New Delhi, India) prior to the closure of that laboratory. The current unavailability of glycosylated Dro, combined with the difficulty of its synthesis, precluded its inclusion in the present work.

Also, the authors look at inhibition of activity (toeprint) and reporter assays to report on “activity” of these peptides. But what about affinity? Could the differences simply be because of altered affinities of these different peptides (and the chimeras discussed later on in the paper) to the ribosome and that activity can be restored if one adds the amount of peptide that is well above their K_d ? It seems like this is a missing part of the story especially in light of the potentially conflicting data.

As noted above, we do not view our data as “potentially conflicting”; rather, we consider them fully consistent with the binding modes and mechanisms of translation inhibition exhibited by sequence-similar Type I and Type II PrAMPs. While we agree that determining the affinity of PrAMPs for the ribosome could yield useful insights, we chose not to pursue such measurements for several reasons:

First, a simple K_d determination would not distinguish between different binding orientations (direct versus inverted). Second, commonly used approaches to measure PrAMP–ribosome affinity require modifying the peptide structure by adding a fluorescent tag. Such modifications may affect not only the measured affinity but also the binding mode itself, which is central to our study, thereby complicating data interpretation. Third, our biochemical and structural results show that sequence-similar PrAMPs can bind the ribosome in either direct or inverted orientations, and some peptides can even adopt both modes (Fig. 4). Although increasing PrAMP concentration well above the K_d would increase ribosome occupancy, it would not alter the distribution between binding modes, which is the key focus of our work.

The authors mention that in prior studies of Dro bound to ribosome, they show two populations of Dro where they differed in the orientation of Dro (the authors reference 36). To the best of my knowledge, this is not stated in that publication so this is surprising. It does raise concern that if Type II peptides can adopt both positions, perhaps this means that it is not so surprising that they find Dal2 is in an inverted position? This point is confusing.

The reviewer is mistaken. The previous study (ref. 36) did NOT show binding of Dro in “different orientations”, but rather showed that binding of Dro in the SAME (Type II-like) orientation may arrest a fraction of the ribosomes at the start codon. Specifically, we write in the current paper “Our previous cryo-EM analysis of the ribosome complexed with Dro showed a minor subpopulation of particles with the PrAMP in the NPET in the classic Type II orientation (Fig. 1c, d) but bound to the initiating, not the terminating, ribosome, presumably inhibiting the first act of translocation”. This is exactly the reason why in the current work we have carried out extensive structural studies to demonstrate that some of the Dro-like peptides (in contrast to Dro!) inhibit initiation of translation by binding with an inverted orientation.

The authors next test chimeric DalDro and DroDal peptides to determine which part of the N or C terminus are important for activity. While DroDal is no longer active, DalDro changes from a Type II to a Type I based upon toeprint analysis. Since the authors used a reporter assay for prior studies, it makes sense to also test these chimeras in this assay.

A general translation inhibition assay, where the mode of PrAMP action can be deduced only from the difference in inhibition efficiency, provides less precise information about the mode of the PrAMP action than the toeprinting analysis which shows whether the ribosome is arrested at the start or stop codon. General inhibition of in vitro translation assay (Fig. 2c) was initially used as an indirect confirmation of the unexpected toeprinting results and was done prior to the structural studies that provided a solid foundation for our biochemical findings. Because the mode of DalDro binding and action has been verified biochemically (toeprinting) and structurally (cryo-EM), we believe including an additional indirect verification would be superfluous.

The interactions between DalDroS and 23S rRNA are shown in Fig 5e and 5f. Maps should be shown for both (I believe surfaces are shown in Fig 5f).

The density maps have been added to panels e and f of Fig. 5 as requested.

Fig 6 shows a model for how these peptides engage the NPET and L4 and L22 r-proteins are shown but are never discussed in this manuscript despite the fact that the C-terminus of Dal2 seems to clearly interact with L4 and L22 (Fig 6a). This needs to be discussed.

We do not observe direct interaction between the PrAMPs and L4 or L22, they are only included in the Fig 6 for reference since they are in close proximity to the region of the PrAMPs that extends into the tunnel.

The last sentence of the Intro is intriguing in that the authors imply that flipping the orientation of the PrAMP could be a strategy by insects to combat bacterial pathogens. This is never expanded on anywhere else in the manuscript. Is this known? Or would insects just encode both Type I and Type II PrAMPs and not rely upon costly mutations?

We did expand on this point in the Discussion. Specifically, we wrote:

“The narrow sequence boundaries between Type I and Type II PrAMPs open attractive evolutionary opportunities for the host. As translation-inhibiting antibiotics, the termination-arresting Type II PrAMPs offer a general advantage of not only arresting the peptide-bound ribosome at the stop codon of an ORF, but also taking offline all the trailing ribosomes on the same mRNA {Mangano, 2020 #9445}. However, the action of Type II PrAMPs depends on the properties of the RFs that could be easily mutated to render bacteria resistant to the PrAMP {Florin, 2017 #8486;Baliga, 2021 #9598}. In contrast, the action of Type I PrAMPs is RF-independent and resistance cannot be easily achieved by target site mutations {Gagnon, 2016 #8179}. If the host is infected by a new or evolved pathogen that is impervious to the action of Type II PrAMPs, it is only a few or even one mutation away from converting the Type II PrAMP to a Type I whose action would not be affected by the properties of the pathogen’s RFs. Thus, not only the small size of the PrAMPs, but also the ability to switch their mode of binding and action by a minimal number of mutations could benefit the host in the ever-going race with the evolving bacterial pathogens. “

Following the reviewer’s comment, in the revised manuscript we added an additional sentence saying:” Noteworthy, insects often encode multiple isoforms of PrAMPs that differ in just a few amino acids. It might be a strategy for generating a repertoire of PrAMPs, with some capable of flipping, if the need should arise.”

Minor-p12 typo (whreas)

Thank you for catching this. The typo has been corrected.

Reviewer #4 (Remarks to the Author):

This is a very comprehensive study demonstrating how a small number of mutations can alter—or even flip—the binding mode of PrAMPs within the bacterial ribosome.

We thank the reviewer for the favorable evaluation of our study.

There are a few questions I would like the authors to address.

- The work focuses on Drosocin variants in *E. coli*. Are there mechanistic insights that could guide rational mutant design to modulate or flip binding modes across different bacterial species and across distinct PrAMP families, which may exhibit sequence and structural diversity?

We know from our unpublished studies that the mode of PrAMP action may be critically affected by the properties of the release factors as well as by the structure of the nascent peptide exit tunnel in ribosomes of different bacterial species. However, the currently available structures of the ribosome/PrAMP complexes are limited to *E. coli* (with only two additional structures of Type II PrAMPs bound to the *Thermus thermophilus* ribosome and one Type I structure on a *Vibrio natriigen* ribosome). Unfortunately, this information is insufficient to make any reliable predictions of what alterations in the structure of the known PrAMPs would lead to flipping of their binding pose and changing the mode of action.

- Additionally, do the reported mutations influence membrane uptake efficiency? Since intracellular transport (e.g., via SbmA) is essential for PrAMP activity, it would be important to clarify whether the mutations affect ribosome binding exclusively or also alter cellular entry, thereby contributing to differences in antimicrobial activity.

The reviewer is absolutely correct that small changes in the structure may affect the PrAMP's uptake as well as their susceptibility to proteases (see, for example, PMID 23114765, 24164266). These factors have complicated interpretation of the mechanisms of action of PrAMPs in some of the previous studies. This is the reason, why for our mechanistic studies we specifically chose to carry out our biochemical experiments in the cell-free translation system. It is also important to keep in mind that it is largely unknown which bacterial species are targeted

by the insect-encoded PrAMPs in nature. Therefore, in our view, the effect of PrAMPs mutations on uptake by *E. coli* cells would have little impact of our understanding of the antibacterial properties of specific PrAMPs either in nature or in clinic. Furthermore, previous studies of PrAMP's uptake involved peptides modified with the fluorescence tags that can alter the chemical and mechanistic properties of the PrAMP. For these reasons, we chose to stay away from the uptake and stability issues, focusing specifically on the on-target activity of the PrAMPs.

- Can the equilibrium between flipped orientations (e.g., observed for DalDro) be characterized quantitatively to support the proposed near-equivalent binding energies (e.g., KD values)?

Like the previous questions of the Reviewer 4, this is also an important question, which, however, is very difficult to address experimentally. The binding of the peptides to the ribosome have been previously studied using competition with the radiolabeled ribosome-targeting antibiotics or employing changes in fluorescence polarization of the fluorescently labeled peptides. None of these approaches can distinguish between the binding mode of the peptide in the exit tunnel. Furthermore, radioactive antibiotics have become prohibitively expensive, whereas attaching of a fluorescence tag can significantly alter the chemical nature of the peptide which, as our results reported in this study show, can affect the binding mode of the PrAMP. For these reasons, we are unable to carry out these interesting, but very difficult experiments.